# Immune Checkpoint Restoration as a Therapeutic Strategy to Halt Diabetes-Driven Atherosclerosis

**DOI:** 10.3390/biology14121731

**Published:** 2025-12-03

**Authors:** Dwaipayan Saha, Preyangsee Dutta, Abhijit Chakraborty

**Affiliations:** 1Metabolic, Nutrition and Exercise Research (MiNER) Laboratory, The University of Texas at El Paso, El Paso, TX 79968, USA; dsaha2@miners.utep.edu (D.S.); pdutta@miners.utep.edu (P.D.); 2Department of Investigational Cancer Therapeutics (ICT), The University of Texas MD Anderson Cancer Center, Houston, TX 77030, USA

**Keywords:** PD-1, CTLA-4, diabetes, atherosclerosis, T cells, immune checkpoints, cardiovascular immunology, immunotherapy, vascular inflammation

## Abstract

People living with diabetes face a much greater risk of heart disease, not only because of high blood sugar but also because their immune system becomes unbalanced in ways that harm the blood vessels. Under healthy conditions, the body uses natural “immune brakes” called immune checkpoints, to prevent unnecessary inflammation. In diabetes, these protective pathways weaken, allowing immune cells to become overly active and damage vessel walls. This accelerates the development of fatty plaques that can block arteries, increasing the risk of heart attack and stroke. In this review, we explain what happens to these immune brakes in both major forms of diabetes, why their failure leads to faster and more dangerous artery disease, and how restoring their function might provide a new way to protect the heart. Early research suggests that strengthening these checkpoints can calm harmful inflammation and help stabilize plaques before their rupture. Understanding how immunity, metabolism, and blood vessel health interact in diabetes offers a clearer path toward improved risk prediction and more precise treatments. Repairing this immune imbalance may help reduce the heavy burden of cardiovascular disease in patients with diabetes.

## 1. Introduction

Diabetes is a global health crisis that affects over 415 million people and is projected to affect 642 million by 2040 [1]. Over 75% of individuals with diabetes live in low- and middle-income countries that lack advanced cardiovascular care. Cardiovascular disease is the leading cause of death among individuals with diabetes, with atherosclerotic complications responsible for over 32.2% of these deaths, resulting in nearly four million premature fatalities every year [2]. The diabetic vasculature represents a uniquely hostile microenvironment in which persistent hyperglycemia, insulin resistance, and dyslipidemia lead to chronic inflammatory activation that fundamentally accelerates atherogenesis. While decades of research have defined glycation, oxidative stress, and lipotoxicity as central metabolic drivers, the immune mechanisms that orchestrate vascular inflammation in diabetes remain unknown. Recent research has demonstrated that hyperglycemia-driven metabolic stress directly impairs immune checkpoint pathways, including PD-1, PD-L1, CTLA-4, TIM-3, and LAG-3, by altering T-cell bioenergetics, mitochondrial function, and redox balance [3]. Complementary evidence from chronic inflammatory and cardiovascular disease models shows progressive suppression of PD-1/PD-L1 and CTLA-4 signaling, supporting the concept that diabetes represents a systemic immune-checkpoint–impaired state [4].

The discovery of immune checkpoint proteins represents a major shift in tumor immunity research. Two major proteins, programmed cell death protein 1 (PD-1) and cytotoxic T lymphocyte-associated protein 4 (CTLA-4), function as master controllers of T cell activation, maintaining immune homeostasis while determining the fate of anti-tumor responses. From an oncological perspective, therapeutic blockade of these inhibitory signals unleashes potent antitumor immune responses, transforming cancer treatment [5]. However, this success has revealed a clinically significant paradox, whereby the systematic disruption of checkpoint signaling can precipitate profound cardiovascular toxicity, including accelerated atherosclerosis, myocardial infarction, and stroke [6,7].

PD-1 and CTLA-4 function as sequential brakes on T cell activation, with detailed mechanisms discussed in the following sections [8]. Chronic hyperglycemia impairs checkpoint expression via multiple pathways [9].

Interestingly, large-scale meta-analyses of patients with cancer receiving immune checkpoint inhibitors demonstrated a striking 3.4-fold increase in major adverse cardiovascular events, with the highest risk observed in patients with pre-existing diabetes or metabolic syndromes [10].

Following immune checkpoint inhibition, the occurrence of fulminant atherosclerotic progression in stable coronary lesions has become a growing concern, suggesting that checkpoint pathways actively constrain plaque development and maintain lesion stability. Conversely, patients with diabetes exhibit significantly reduced immune checkpoint expression, with the degree of reduction corresponding to glycemic control, inflammatory burden, and cardiovascular risk scores [11].

Several innovative therapeutic strategies have emerged to restore checkpoint function in the diabetic vasculature [12]. PD-L1 agonistic antibodies, designed to enhance rather than block PD-1/PD-L1 interactions, have demonstrated remarkable efficacy in preclinical atherosclerosis models by reducing plaque burden and promoting lesion stability [13]. CTLA-4 mimetic compounds show benefits similar to those that target the inflamed endothelium [14]. Most intriguingly, gene therapy approaches using adenoviral vectors to restore checkpoint molecule expression directly within vascular tissues offer the potential for durable, localized immunomodulation without systemic immunosuppression [15].

This review synthesizes a single, testable translational framework that bridges islet autoimmunity and diabetic vascular disease. We show how autoimmune checkpoint failure in Type 1 and metabolism-driven checkpoint suppression in Type 2 converge on a common metabolic–epigenetic axis that sustains the loss of PD-1/PD-L1, LAG-3, and related regulators following the initial trigger, explaining persistent inflammation across both islet and vascular compartments. Using this mechanistic framework, we linked tissue-resident memory T cell biology and circadian control of checkpoint expression to explain tissue- and time-specific vulnerability. Based on these insights, we propose clinical biomarker panels for patient selection and monitoring, imaging and surrogate endpoints for early trials, time-of-day-based dosing, and endothelium-targeted delivery to restore local immune tolerance while limiting systemic effects. Using recent human single-cell and spatial maps that identify precise cellular niches and measurable pharmacodynamic markers, this review shifts from description to action, offering a pragmatic translational roadmap for evaluating checkpoint restorative therapies to prevent diabetes-accelerated atherosclerosis in the future.

While this review primarily addresses type 2 diabetes, which accounts for 90–95% of cases, checkpoint dysfunction mechanisms also extend to type 1 diabetes through distinct autoimmune pathways, which are discussed in subsequent sections. A conceptual overview of these progressive immune checkpoint disruptions, from healthy homeostasis to diabetic plaque destabilization, is shown in Figure 1.

## 2. Immune Checkpoint Biology and Dual Roles Across Systems

Immune checkpoint regulation operates through distinct molecular circuits that require systematic examination. CTLA-4 and PD-1 represent complementary regulatory systems with unique temporal and spatial frameworks. During early T cell priming, the initiation of CTLA-4 outcompetes the activating receptor CD28 for CD80/CD86 binding on antigen-presenting cells (APCs). This establishes the first layer of immune regulation in our body. PD-1 assumes control during the later phases, dampening ongoing T cell responses in the peripheral tissues. It recruits SHP-1/SHP-2 phosphatases, which systematically shut down key signalling pathways [16,17].

This dual checkpoint architecture creates a two-layer safety system. CTLA-4 prevents inappropriate T-cell activation during the initiation phase. PD-1 constrains tissue-damaging responses once the T cells are activated. In the context of atherosclerosis, both mechanisms provide critical protection against inflammatory and vascular damage. Endothelial cells express PD-L1 to maintain T cell quiescence. Regulatory T cells deploy CTLA-4 to suppress harmful immune responses at plaque sites [18,19]. The functional importance of these pathways is evident from genetic manipulation studies. Knockout of either checkpoint pathway in mouse models results in dramatically accelerated atherosclerosis [20,21].

### 2.1. PD-1 Pathway as the Primary Brake on T Cell Effector Function

PD-1 functions as a molecular switch that governs T cell activation and tissue tolerance [22]. This type I transmembrane glycoprotein, encoded by the PDCD1 gene, is transiently upregulated on CD4^+^ and CD8^+^ T cells following activation, with peak expression occurring in 24–48 h post-stimulation [23]. PD-1 signals through two distinct ligands, PD-L1 (B7-H1/CD274), which is broadly expressed on hematopoietic and non-hematopoietic cells including vascular endothelium and PD-L2 (B7-DC/CD273), which exhibits more restricted expression primarily on antigen-presenting cells [24].

Within the atherosclerotic microenvironment, PD-1 signaling emerges as a critical guardian against excessive vascular inflammation. Endothelial cells constitutively express PD-L1 under homeostatic conditions, creating a protective barrier that prevents aberrant T cell activation within the walls of healthy vessels [25]. During early atherogenesis, oxidized low-density lipoproteins and inflammatory cytokines upregulate endothelial PD-L1 expression through NF-κB and IRF-1 transcriptional pathways, establishing a negative feedback loop that constrains T cell-mediated vascular damage [26]. Genetic ablation of PD-1 in atherosclerosis-prone mouse models results in dramatically accelerated lesion development, characterized by enhanced CD8^+^ T cell infiltration, increased interferon-γ production, and promotion of vulnerable plaque phenotypes with thin fibrous caps and extensive necrotic cores [13].

### 2.2. CTLA-4 Pathway as the Master Regulator of Costimulatory Signaling

CTLA-4 functions as the primary checkpoint that controls the initiation of T cell responses through the competitive inhibition of costimulatory signaling [27]. Unlike PD-1, which primarily regulates ongoing immune responses in peripheral tissues, CTLA-4 operates at the critical interface between antigen-presenting cells and T cells during the priming phase of immune activation [27]. This homodimeric glycoprotein shares structural homology with the costimulatory receptor CD28 but exhibits a 20-fold higher affinity for its shared ligands CD80 (B7-1) and CD86 (B7-2) on dendritic cells, macrophages, and activated B cells [28].

The cell-intrinsic signaling mechanisms of CTLA-4 involve the recruitment of protein phosphatases PP2A and SHP-2, which dephosphorylate important TCR signaling intermediates and promote cell cycle arrest at the G1/S checkpoint [29]. Notably, CTLA-4 is constitutively expressed at high levels on regulatory T cells, where it serves as both a functional suppressor and a lineage-defining marker [30]. Treg-expressed CTLA-4 not only maintains these cells in an anergic state but also enables them to suppress conventional T cell responses through competitive consumption of costimulatory ligands and direct cell-contact-dependent mechanisms [31].

In atherosclerotic contexts, CTLA-4 signaling provides crucial protection against autoimmune-like vascular inflammation [27]. The arterial wall contains abundant antigen-presenting cells, including dendritic cells and macrophages, which present oxidized lipid antigens and other damage-associated molecular patterns to infiltrating T cells. CTLA-4 expression on vascular Tregs serves as a critical brake preventing excessive activation of atherogenic Th1 and Th17 responses [32]. Experimental blockade of CTLA-4 in atherosclerotic mice precipitates severe exacerbation of plaque inflammation, characterized by massive T cell infiltration, cytokine storm-like local inflammation, and accelerated progression to advanced lesions [33]. Conversely, enhancement of CTLA-4 through agonistic approaches promotes plaque stabilization and reduces inflammatory gene expression within vascular tissues.

## 3. Diabetic Atherosclerosis: An Immune Checkpoint-Impaired State

The checkpoint regulatory networks that normally safeguard vascular homeostasis become progressively impaired in the diabetic milieu, transforming the arterial wall into a site of unchecked immune activation [34]. This pathological transformation represents a fundamental departure from the balanced immune surveillance that characterizes healthy vasculature.

### 3.1. Hyperglycemia-Mediated Disruption of Checkpoint Networks

A chronic hyperglycemic state orchestrates a multifaceted assault on immune checkpoint integrity through interconnected metabolic and inflammatory pathways. Persistent glucose elevation fundamentally alters T cell metabolism, forcing a shift toward aerobic glycolysis that impairs the oxidative phosphorylation required for optimal PD-1 expression and function [35]. This metabolic reprogramming disrupts the delicate balance between glycolytic and oxidative phosphorylation pathways, with hyperactivated mTOR signaling promoting effector T cell differentiation while suppressing checkpoint molecule upregulation [36]. Simultaneously, advanced glycation end products (AGEs) serve as potent danger signals that activate pattern recognition receptors on antigen-presenting cells, with AGE engagement of RAGE triggering robust NF-κB activation that promotes cellular maturation while paradoxically downregulating checkpoint ligands CD80, CD86, PD-L1, and PD-L2 [37,38]. Similar dynamics also occur in vascular endothelium with glucose activating protein kinase C, initiating signaling cascades that suppress PD-L1 while enhancing pro-inflammatory gene transcription [39].

Oxidative stress is elevated by mitochondrial dysfunction and NADPH oxidase activation, directly impairing checkpoint signaling by oxidizing critical cysteine residues within the PD-1 receptor and activating transcriptional repressor FOXO1, which silences the PDCD1 promoter [40]. The dyslipidemic profile characteristic of diabetes further disrupts checkpoint function by altering T cell membrane lipid raft composition, impairing the spatial organization of checkpoint receptors and uncoupling receptor engagement from inhibitory signaling [41].

Emerging evidence demonstrates a dose-dependent relationship between glycemic control and immune checkpoint integrity, linking metabolic status to vascular immune homeostasis beyond binary diabetic classifications. Sustained hyperglycemia (HbA1c > 8.5%) is associated with progressive PD-1 downregulation on T cells, weakening inhibitory signaling and permitting vascular immune activation [42,43]. Parallel changes occur in the vascular endothelium, where PD-L1 expression on arterial endothelial cells declines proportionally to rising glucose levels, with patients having HbA1c levels > 9.0% exhibiting near-complete loss of PD-L1, thus promoting immune cell infiltration and vascular inflammation [25]. Disease duration further modulates this effect, and patients with <5 years of diabetes retain partial checkpoint function even under poor control, whereas those with >15 years exhibit severe dysfunction even at moderate HbA1c levels, underscoring the cumulative burden [44]. Mechanistically, glucose levels > 140 mg/dL suppress transcription factors (FOXO1, NFAT) essential for PD-1 while activating mTOR-driven effector T cell expansion. Advanced glycation end products rise exponentially above 180 mg/dL, further amplifying checkpoint suppression [45,46]. Regulatory T cells show selective vulnerability, with CTLA-4 expression sharply reduced as HbA1c exceeds 8.0%, leading to disproportionate loss of suppressive function [47]. Importantly, strict glycemic control appears protective with patients maintaining HbA1c levels < 7.5% preserving checkpoint function more effectively than those >8.0%, showing benefits extending to cardiovascular protection and immune regulation [48]. Unfortunately, complete recovery may not be achievable [49]. Immune checkpoint profiling could refine risk stratification and guide targeted interventions in patients with subclinical immune dysregulation despite acceptable HbA1c levels [50].

In contrast, hyperglycemia suppresses immune checkpoint function, whereas metabolic reprogramming and epigenetic remodeling impair T cell regulation in diabetes. While direct evidence linking these epigenetic mechanisms to checkpoint dysfunction in human diabetic atherosclerosis remains limited, converging data from cancer immunology and chronic inflammatory disease models provide strong mechanistic rationale. Impaired fatty acid oxidation (FAO), a critical pathway for memory T cell differentiation and sustained CTLA-4 expression, becomes markedly compromised in diabetes due to disrupted lipid metabolism and mitochondrial dysfunction, thereby diminishing T cell longevity and checkpoint upregulation [51]. At the molecular level, diabetes causes significant shifts in metabolic cofactors, including NAD^+^, α-ketoglutarate, and acetyl-CoA, which disrupt the normal activity of chromatin-modifying enzymes [52]. These alterations may render long-standing dysfunction resistant to glucose-lowering therapies, highlighting the need for DNMT inhibitors, HDAC inhibitors, or TET activators as corrective strategies. Specifically, reduced α-ketoglutarate to succinate ratios impair TET-mediated DNA demethylation, while NAD^+^ depletion compromises SIRT1-dependent histone deacetylation [53]. Lipid transport mechanisms add additional complexity to this dysfunction. CD36-mediated fatty acid uptake drives metabolic reprogramming and immune evasion. Evidence from chronic inflammatory diseases and gastric cancer underscores the broader role in immunopathology [54,55].

Alterations in amino acid metabolism provide additional mechanisms of dysfunction. Limited glutamine and arginine availability reshapes mTOR signaling, thus raising activation thresholds and restricting checkpoint expression through metabolic constraints [56]. Beyond acute metabolic stress, the emerging concept of metabolic memory suggests that prior hyperglycemia may imprint checkpoint gene loci, such as PDCD1 and CD274, with persistent epigenetic marks, although direct demonstration in vascular tissues is required for validation [57]. The epigenetic mechanisms described above are well established in the context of cancer and autoimmune diseases. However, their specific roles in diabetic atherosclerotic plaques require direct validation through tissue-specific chromatin profiling and functional studies. The proposed metabolic–epigenetic axis represents an integrative framework synthesizing existing knowledge that requires experimental confirmation.

Epigenomic profiling is a critical research priority for linking checkpoint accessibility to clinical outcomes and validating these mechanisms in human diabetic vascular disease.

Interferon-γ dominates the cytokine landscape within diabetic atherosclerotic lesions. Th1 cells become the predominant T cell subset in these plaques, secreting interferon-γ at levels sufficient to shift macrophages toward M1 polarization and accelerate oxidative modification of lipids [58]. Immune cells experience reduced checkpoint receptor expression, whereas vascular tissues lose their capacity for ligand presentation. IL-17A production by Th17 cells adds another layer of dysfunction. This cytokine stimulates chemokine secretion and upregulates matrix metalloproteinases, which compromise plaque stability [59]. This oxidative environment activates transcriptional repressors that silence checkpoint genes [60]. These observations demonstrate that checkpoint failure in diabetes reflects a convergent assault from metabolic derangement and inflammatory activation, rather than a single pathway disruption.

### 3.2. Pro-Atherogenic T Cell Polarization in Checkpoint-Deficient Environments

The systematic depletion of checkpoint control creates permissive conditions for pathogenic T cell responses that drive accelerated atherogenesis. In the absence of adequate PD-1-mediated restraint, CD4^+^ T cells undergo preferential differentiation toward pro-inflammatory Th1 and Th17 phenotypes [61]. This enhancement is mechanistically driven by enhanced STAT1 and STAT3 signaling, leading to expression of T-bet and RORγt transcription factors [62].

Th1 cells, characterized by high interferon-γ production, become the dominant subset within diabetic atherosclerotic plaques, directly activating macrophages toward the M1 phenotype and accelerating lipid oxidation while promoting smooth muscle cell apoptosis [63]. Th17 cells contribute through IL-17A production, stimulating chemokine release and upregulating matrix metalloproteinases that destabilize plaques [64].

The checkpoint-deficient environment also affects CD8^+^ T cell responses, where impaired oxidative phosphorylation capacity reduces their ability to maintain exhausted states. These cytotoxic cells undergo clonal expansion with enhanced effector functions, directly inducing apoptosis in vascular smooth muscle cells and endothelial cells through granzyme B and perforin-mediated mechanisms [65].

### 3.3. Treg Dysfunction Driving Immune Imbalance

Checkpoint dysfunction involves the systematic impairment of regulatory T cell function, representing a complete breakdown of the primary suppressive mechanism of the immune system. Hyperglycemia creates a metabolic environment that fundamentally opposes Treg survival and function, as these cells uniquely depend on robust mitochondrial oxidative phosphorylation and fatty acid oxidation to maintain their suppressive capabilities, whereas diabetes favors aerobic glycolysis over oxidative metabolism [66]. Transcriptomic analyses have revealed a 60% reduction in CTLA-4 expression in diabetic atherosclerotic plaques, occurring through both transcriptional downregulation and microRNA-mediated mRNA degradation [67]. These functional consequences extend beyond simple numerical deficits to include dramatically reduced capacity for trans-endocytosis of CD80/CD86 costimulatory molecules and severely impaired production of immunosuppressive enzymes that normally maintain vascular immune tolerance [68].

### 3.4. Endothelial Checkpoint Dysfunction Marks a Breach in the Vascular Immune Barrier

The diabetic endothelium undergoes alterations in checkpoint ligand expression, compromising its protective barrier function. Glucose-induced activation of protein kinase C triggers inflammatory signaling pathways that paradoxically suppress PD-L1 expression while upregulating pro-inflammatory gene expression [69]. Advanced glycation end products amplify this dysfunction by generating reactive oxygen species that oxidatively modify the transcription factors required to maintain PD-L1 expression, effectively silencing the checkpoint ligand at the vascular interface, where immune regulation is most crucial [58].

### 3.5. Checkpoint Dysfunction as a Driver of Plaque Vulnerability in Integrated Pathophysiology

The convergence of T cell checkpoint dysfunction, regulatory T cell impairment, and endothelial barrier breakdown creates immune dysregulation that accelerates atherosclerotic progression and promotes plaque instability [70]. The checkpoint-deficient diabetic plaque exhibits enhanced T cell infiltration, especially interferon-γ-producing Th1 cells and cytotoxic CD8^+^ T cells, creating a pro-inflammatory microenvironment with large necrotic cores, thin fibrous caps, and extensive inflammatory infiltration [71].

This mechanistic understanding provides a rationale for therapeutic strategies aimed at restoring checkpoint function and correcting the underlying oxidative phosphorylation deficits, representing a transformative approach toward targeted immunomodulation as a cornerstone of diabetic cardiovascular care. These interconnected molecular and systemic mechanisms of immune checkpoint dysregulation in diabetes are shown in Figure 2. While human studies have established an association between checkpoint dysfunction and cardiovascular risk, the proposed framework establishes causality through experimental validation. Pharmacologic blockade of CTLA-4 in hyperlipidemic mice produced nearly a two-fold increase in atherosclerotic lesion area accompanied by enhanced T cell infiltration and macrophage accumulation in plaques, whereas CTLA-4 overexpression in apolipoprotein E-deficient mice reduced atherosclerotic lesion formation by approximately 35% and decreased macrophage and CD4^+^ T cell accumulation in the aortic root [72]. Additionally, PD-1/PD-L1 deficiency aggravated atherosclerosis in LDL receptor-deficient mice, increasing CD4^+^ and CD8^+^ T cell numbers within lesions, while agonistic PD-1 antibody treatment reduced atherosclerosis development, decreased IFNγ-producing CD4^+^ T cells by 23%, and increased atheroprotective IL-10-producing T cells by 47% [13]. Cell-specific manipulations in endothelial cells and regulatory T cells identify the vascular and immune compartments as mechanistic mediators. Temporal analyses of diabetic models have demonstrated that checkpoint loss precedes plaque progression. These molecular findings distinguish mechanistic causality from epidemiological correlations.

## 4. Clinical Translation and Human Evidence for Checkpoint Dysfunction in Diabetic Atherosclerosis

Growing evidence from human studies confirms that immune checkpoint dysfunction plays a meaningful role in diabetic cardiovascular disease, validating laboratory findings in real patients [34]. These investigations provide crucial validation that experimental observations translate meaningfully to human pathophysiology, establishing immune checkpoint impairment as a clinically significant contributor to accelerated cardiovascular disease in diabetes. Preclinical studies have consistently shown that genetic or pharmacological restoration of PD-1 and CTLA-4 signaling reduces vascular inflammation and plaque formation. However, these findings are limited by heterogeneity and short intervention periods. Most animal studies use young, male, hyperlipidemic mice under acute inflammatory conditions that partly mimic the chronic metabolic dysfunction observed in human diabetes [73]. Clinical evidence linking immune checkpoint inhibition to cardiovascular events is strong [74], but largely observational and confounded by cancer-related inflammation and therapy exposure. No interventional trial has tested checkpoint agonists for atherosclerotic disease, and the long-term safety of systemic checkpoint activation remains unknown. The clinical data support checkpoint restoration as a potential therapeutic approach; however, translating this to human therapy will require controlled dose-finding studies, vascular-specific agonists, and long-term safety data. Key clinical studies highlighting checkpoint dysfunction in diabetic cardiovascular diseases are summarized in Table 1.

## 5. Challenges in Translation and Therapy

Immune checkpoint dysfunction in diabetic atherosclerosis remains compelling but faces critical uncertainties that must be addressed before it can be clinically translated. These challenges focus on safety, precision of targeting, and the relevance of laboratory findings to patient outcomes. A comparative overview of key immune checkpoints, their vascular roles, diabetes-induced alterations, and therapeutic strategies is presented in Table 2.

### 5.1. Safety Considerations in Systemic Checkpoint Enhancement

Restoring checkpoint functions to reduce vascular inflammation raises critical concerns regarding the balance between therapeutic efficacy and immune safety. The challenge lies in achieving anti-inflammatory efficacy without tipping the immune balance toward suppression. Although preclinical findings indicate possible vascular benefits, most safety concerns are inferred from the cancer immunotherapy field rather than grounded in direct evidence from checkpoint agonist studies, highlighting the need for rigorous mechanistic and translational validation before clinical application. Systemic enhancement of PD-1 or CTLA-4 signaling, while potentially beneficial for suppressing pathogenic vascular immune responses, could precipitate excessive immune suppression with catastrophic consequences for host defense mechanisms [81]. This therapeutic dilemma is acute in elderly diabetic populations, where baseline immune senescence and co-morbid conditions create heightened vulnerability to opportunistic infections and malignancy. These safety concerns remain unresolved in the absence of clinical trials testing checkpoint agonists in diabetic atherosclerosis.

The immunosuppressive potential of checkpoint enhancement extends beyond infectious disease susceptibility to encompass impaired tumor immunosurveillance, a concern of paramount importance [82]. The intricate relationship between checkpoint pathways and anti-tumor immunity suggests that restoring these inhibitory signals may protect the vasculature while increasing the risk of cancer [83]. The concerns about systemic immune suppression are reasonable, but PD-1/PD-L1 agonism in diabetic atherosclerosis differs from broad immunosuppressive therapies. These agonists restore the physiological checkpoint signaling that is downregulated in the diabetic vasculature, rather than globally suppressing immune function. Experimental studies have shown that partial PD-L1 agonists normalize local immune-endothelial interactions and reduce vascular inflammation without impairing systemic antiviral or antibacterial responses [84,85]. Vascular-targeted delivery platforms further limit systemic exposure. VCAM-1 or ICAM-1-conjugated nanoparticles and drug-eluting stents confine checkpoint engagement to the arterial wall, reducing infection risk [86]. These mechanisms suggest that PD-1/PD-L1 agonism represents localized immune recalibration rather than generalized suppression, a distinction that is important in patients with diabetes with baseline immune deficits.

### 5.2. Vascular Targeting and Delivery Challenges

Current therapeutic approaches for checkpoint modulation suffer from fundamental limitations in tissue specificity, relying on systemic delivery strategies that lack the precision required for selective vascular intervention [87]. The development of vascular-targeted delivery platforms represents a critical frontier in the translation of checkpoint-based therapeutics from experimental concepts to clinical reality. Sophisticated nanotechnology approaches, including lipid nanoparticles engineered with endothelial-specific ligands and polymer-based drug delivery systems, offer promising avenues for achieving vascular selectivity while minimizing systemic exposure [88].

Device-based delivery strategies present complementary opportunities for localized checkpoint modulation, particularly through advanced stent technologies capable of controlled drug release. These platforms could theoretically deliver checkpoint agonists directly to sites of vascular injury or inflammation, achieving therapeutic concentrations within the arterial wall while avoiding systemic immunosuppression [89]. However, the bioengineering challenges inherent in developing such systems, including drug stability, release kinetics, and biocompatibility, require extensive preclinical validation before clinical translation becomes feasible. The safety evaluation of vascular-targeted checkpoint delivery is critical, as localized platforms can alter biodistribution, retention, and immune activation thresholds, with off-target accumulation in the liver, spleen, or lymphoid tissues potentially disrupting systemic immunity [90]. Hydrogel and nanoparticle depots delivering checkpoint inhibitors require careful assessment of local inflammation, thrombogenicity, degradation kinetics and duration of receptor engagement to ensure therapeutic efficacy without excessive immunosuppression [91]. Robust pharmacokinetic and pharmacodynamic characterization including imaging, circulating drug monitoring and PET-based tracking is essential before clinical implementation of these vascular-targeted systems [92].

### 5.3. Bridging the Longitudinal Evidence Gap in Checkpoint Research

A major translational gap persists between preclinical research and clinical application. Despite the mechanistic insights from experimental studies, most published evidence is derived from isolated cell cultures or single-time-point animal models. Human data are sparse and predominantly cross-sectional. This fundamental limitation severely constrains our ability to translate bench discoveries into bedside interventions for patients with diabetes.

The mechanistic framework presented in Table 1 illustrates the disparities between experimental and clinical evidence. However, longitudinal clinical trials examining PD-1/PD-L1 signaling in patients with diabetes and cardiovascular outcomes remain limited. Durable checkpoint restoration in diabetic arteries has not been previously studied. Any effort to restore checkpoint function must also address these potential risks. Enhancing PD-1 or CTLA-4 activity may weaken tumor immunosurveillance, particularly in patients with latent or undiagnosed malignancies, as forced PD-1 expression has been shown to promote tumor cell proliferation through SHP2–Ras–MAPK signaling [93]. Checkpoint enhancement may also increase vulnerability to infections, since PD-1 signaling is required to limit immune-mediated injury during chronic viral infections [94], and PD-L1 contributes to the control of type I interferon responses in stressed or infected cells [95]. These safety concerns support a cautious, stepwise approach with controlled dosing and close immune monitoring before applying checkpoint-restoring strategies for vascular diseases. Current evidence relies largely on cross-sectional comparisons between diabetic and non-diabetic cohorts, offering snapshots of immune dysfunction rather than a dynamic view of disease progression [96]. Longitudinal studies tracking immune checkpoint evolution across diabetes duration could transform our approach to prevention and treatment. Such investigations might reveal whether intensive glycemic control in early diabetes preserves checkpoint function, potentially preventing the immune cascade that drives accelerated atherosclerosis [97]. These studies could also identify patients experiencing rapid checkpoint deterioration, enabling targeted intervention before irreversible vascular damage occurs. To address these critical knowledge gaps, future research should prioritize longitudinal study designs incorporating immune checkpoint assessment, advanced cardiovascular imaging, and comprehensive clinical phenotyping.

### 5.4. Clinical Management of Cancer Patients with Diabetes

The expanding utilization of immune checkpoint inhibitors in oncology creates urgent clinical dilemmas regarding cardiovascular monitoring and prophylaxis in patients with diabetes and cancer [98]. These individuals face the convergent risks of checkpoint inhibitor-induced cardiovascular toxicity superimposed upon pre-existing diabetic vascular disease; however, evidence-based guidelines for managing this high-risk population remain conspicuously absent. The absence of standardized cardiovascular surveillance protocols leaves clinicians to navigate complex risk–benefit decisions without adequate evidence to guide optimal care.

The heterogeneity of checkpoint inhibitor regimen, including single-agent versus combination therapies and varying dosing schedules, further complicates risk stratification and monitoring strategies. Recent pharmacovigilance analyses suggest that cardiovascular events may occur across a broad temporal spectrum, from acute presentations within weeks of treatment initiation to delayed manifestations months after therapy is completed [99]. This temporal variability underscores the need for comprehensive cardiovascular assessment protocols that extend beyond traditional pre-treatment evaluation to encompass longitudinal monitoring throughout the treatment continuum and beyond.

### 5.5. Genetic Determinants and Personalized Medicine

Individual genetic variations in checkpoint pathways represent an untapped opportunity for personalized cardiovascular care in patients with diabetes. Common variants within PDCD1, CD274, CTLA4, and related genes likely influence baseline checkpoint function, creating differences in atherosclerotic risk and treatment response that could guide precision therapeutic strategies [85]. Individual genetic variations in checkpoint genes, such as PDCD1 (rs11568821 and rs2227981) and CTLA4 (rs231775 and rs3087243), alter receptor expression and signaling thresholds, influencing autoimmune susceptibility and cardiovascular inflammatory activity [100,101]. In diabetes, these functional variants may amplify hyperglycemia-induced checkpoint suppression, creating heterogeneity in vascular inflammation and identifying patients who may differentially benefit from checkpoint-restoring therapies [102].

Large-scale genomic studies examining checkpoint gene variants and cardiovascular outcomes in diabetic populations could identify predictive biomarkers for accelerated atherosclerosis, therapeutic responses, and adverse events [103]. Integrating pharmacogenomic data with clinical and molecular biomarkers offers the potential to develop comprehensive risk prediction models, enabling more precise patient selection and treatment optimization [104]. This genomic approach enables personalized cardiovascular risk stratification and tailored interventions, transforming diabetic cardiovascular care from universal protocols to precision medicine.

### 5.6. Therapeutic Window and Dosing Considerations

The optimal therapeutic approach for checkpoint modulation in diabetic atherosclerosis remains undefined, with critical questions surrounding dose selection, treatment duration, and monitoring strategies [105]. Unlike cancer immunotherapy, where checkpoint inhibition seeks maximal immune activation, cardiovascular applications require more nuanced modulation to restore physiological immune balance without compromising host defense [106]. This therapeutic approach requires a precise understanding of dose–response relationships and real-time biomarkers for monitoring efficacy and safety.

The temporal dynamics of checkpoint dysfunction in diabetic atherosclerosis further complicate therapeutic planning [76]. Early intervention during the initial checkpoint impairment phase may prevent atherosclerotic acceleration, whereas delayed treatment may focus on stabilizing existing plaques or preventing acute events [107]. These considerations highlight the need for comprehensive natural history studies that characterize the evolution of checkpoint dysfunction across the spectrum of diabetic cardiovascular disease.

### 5.7. Regulatory and Implementation Challenges

The translation of checkpoint-based therapeutics for diabetic atherosclerosis faces substantial regulatory hurdles that reflect the novel nature of immunomodulatory approaches for cardiovascular diseases [108]. Current regulatory frameworks, designed primarily for traditional cardiovascular treatments, inadequately address the unique considerations inherent to immune checkpoint modulation [109]. The development of appropriate clinical trial designs, endpoint selection, and safety monitoring protocols requires close collaboration between investigators, regulatory authorities, and clinical communities to establish evidence standards that ensure patient safety while facilitating therapeutic innovations.

The implementation of these technologies in healthcare systems presents equally complex challenges. The specialized expertise required for immune checkpoint biology and adverse event management may not be readily available in all clinical settings [110]. Considerations of cost-effectiveness, provider education needs, and patient selection further complicate its adoption [111]. Addressing these multifaceted challenges through coordinated research initiatives, regulatory engagement, and clinical collaboration is essential to realize the therapeutic potential of immune checkpoint modulation in diabetic atherosclerosis while ensuring patient safety and optimal clinical outcomes.

## 6. Future Directions in Translating Checkpoint Biology to Clinical Practice

Emerging insights into immune checkpoint dysfunction in diabetic atherosclerosis provide a foundation for therapeutic innovation with the potential to reshape cardiovascular risk management [112]. Progress in this area will depend on advances in risk stratification, imaging technologies, and pharmacological strategies that bridge mechanistic understanding with clinical application.

### 6.1. Immune Checkpoint-Based Risk Stratification

Conventional cardiovascular risk models often fail to capture the immune drivers of accelerated atherosclerosis in patients with diabetes. Integrating checkpoint assessment into clinical practice could allow for a more precise identification of inflammatory risk [113]. Flow cytometry profiling of PD-1, PD-L1, and CTLA-4 expression on circulating T cells provides direct insight into their regulatory capacity [114], while soluble checkpoint proteins such as sPD-L1 and sCTLA-4 represent minimally invasive biomarkers. Their incorporation into cardiovascular screening could improve risk prediction in high-risk diabetic populations [115].

### 6.2. Advanced Imaging and Molecular Visualization

Molecular imaging offers new opportunities to link checkpoint biology with the in vivo assessment of atherosclerotic plaques [116]. PET tracers directed against PD-L1 or inflammatory mediators, combined with CT angiography or cardiac MRI, can identify “immunologically active” plaques prone to rupture [117]. Validation in diabetic cohorts is essential, and integration into clinical trials is needed to establish their value in guiding therapy and monitoring response.

### 6.3. Pharmacologic Interactions and Modulation of Checkpoints by Antidiabetic Therapies

Checkpoint dysfunction in diabetes may create opportunities to leverage antidiabetic therapies with potential immunological effects [118]. However, direct evidence demonstrating that these agents restore checkpoint function in patients with diabetes remains limited to observational and preclinical data, allowing the possibility of mechanistic insights requiring prospective clinical validation before rational combination strategies can be implemented.

Metformin, one of the most widely studied medicines for lowering glucose, activates AMPK, enhances fatty acid oxidation, and promotes mitochondrial biogenesis, all of which stabilizes regulatory T cells [119]. These effects align with those of PD-1 or CTLA-4 agonist therapy, while the inhibition of NF-κB and reduction of AGE formation further support checkpoint recovery [120]. Statins provide additional benefits by upregulating PD-L1 on vascular endothelium by reducing T cell activation and promoting regulatory function [121]. In contrast, high-dose insulin may enhance effector T cell activity through glycolytic flux [51] and sulfonylureas can amplify inflammatory signaling via potassium channels [122]. Comprehensive profiling of GLP-1 receptor agonists, SGLT2 inhibitors, DPP-4 inhibitors, and thiazolidinediones across PD-1/PD-L1 and CTLA-4/CD80-86 pathways defining their immunologic signatures [123]. Trial design accounts for pharmacokinetics, sequencing, and circadian regulation of immuno-metabolic pathways [124]. Safety frameworks must be employed when checkpoint modulators are paired with potent anti-inflammatory agents [125]. The therapeutic window for these strategies requires careful dose optimization and biomarker-guided monitoring [126]. However, the successful integration of checkpoint biology with metabolic therapy represents a promising but yet-to-be-validated method that could establish a new potential for personalized immunometabolic medicine, offering individuals with diabetes a novel approach to reduce cardiovascular complications [34].

### 6.4. Longitudinal Immune Surveillance and Dynamic Risk Assessment

Checkpoint pathways in diabetes evolve dynamically with glycemic status, metabolic therapy, and cardiovascular interventions. Longitudinal surveillance is needed to distinguish transient from persistent dysfunction and to define therapeutic windows. Such monitoring could enable adaptive treatment algorithms that adjust in response to changing immune phenotypes rather than relying on fixed baseline measures [127].

### 6.5. Multi-Omics Integration and Predictive Modeling

The heterogeneity of immune dysfunction in diabetic atherosclerosis requires multi-layered analysis. Single-cell sequencing, spatial transcriptomics, proteomics, and metabolomics provide complementary insights, whereas machine learning integration of these datasets can generate predictive immune signatures [128]. These tools may be beneficial for detecting plaque destabilization or therapeutic responses, supporting precision immune–metabolic medicine [129].

### 6.6. Targeted Therapeutic Delivery Platforms

Systemic checkpoint modulation carries risks of broad immunosuppression. Nanoparticles functionalized with endothelial ligands, such as VCAM-1 or ICAM-1, enable selective delivery to inflamed vasculature, whereas bioresorbable hydrogels and drug-eluting stents provide localized, sustained release during revascularization. Nanoparticle platforms conjugated with endothelial-targeting ligands (VCAM-1, ICAM-1) can direct checkpoint agonists to sites of vascular inflammation, whereas bioresorbable hydrogel implants and drug-eluting stent technologies offer controlled, localized immune regulation during revascularization procedures [130]. These approaches require prioritized preclinical validation and early phase clinical testing to be effective. Emerging advances in adoptive cell therapies, such as CAR-T cell engineering, further underscore the potential of precision immunomodulation in cardiovascular disease, drawing on strategies originally developed for oncology [131].

### 6.7. Precision Patient Selection and Therapeutic Optimization

The therapeutic efficacy depends on patient stratification. The early stage of atherosclerotic disease may respond to checkpoint restoration alone, whereas advanced disease may require integration with lipid-lowering, anti-inflammatory, or antiplatelet therapy [132]. Figure 3 summarizes the proposed approach to cardiovascular checkpoint enhancement, advanced drug delivery strategies, and precision patient stratification pathways.

### 6.8. Regulatory Framework and Safety Infrastructure

Clinical translation of checkpoint-based therapies requires stringent regulatory oversight. Trials must extend beyond efficacy to include monitoring of autoimmune events, vaccine responses, and oncologic risk [133]. Collaboration between regulatory agencies, clinical investigators, and immunologists will be essential for developing appropriate trial designs and risk mitigation strategies. Current clinical data on PD-1/PD-L1 agonism remain limited, and this constraint should be acknowledged when considering translational feasibility. Early studies with peresolimab provided short-term safety signals but did not address long-term immune or oncologic outcomes relevant to chronic vascular disease [134]. Similar phase I studies evaluating PD-1 directed fusion molecules, including PF-07209960, reported tolerability and pharmacokinetic profiles but left key questions regarding infection surveillance and sustained immune regulation unresolved [135]. Observations from checkpoint inhibitor trials further illustrate the need for systematic monitoring, as multi-organ immune toxicities including cardiovascular involvement have been documented across several cohorts [136]. Data on vaccine responses in the setting of checkpoint modulation show preserved immunogenicity, although evidence remains incomplete for non-oncologic and metabolically compromised populations [137]. Taken together, these points indicate that checkpoint restoration is a biologically plausible approach, but its therapeutic applicability in diabetes-driven atherosclerosis will depend on dedicated clinical studies that precisely define safety margins with greater precision.

### 6.9. Interdisciplinary Integration and Implementation

The successful translation of checkpoint science into cardiovascular practice demands unprecedented collaboration among cardiologists, immunologists, endocrinologists and vascular biologists. Development of specialized cardio-immuno-metabolic clinics and interdisciplinary research networks will facilitate integration of immune profiling into routine cardiovascular management, while comprehensive educational initiatives must equip healthcare providers with the knowledge to interpret immune biomarkers and implement checkpoint-informed therapeutic strategies [138]. The conceptual trial design incorporating key immunologic and clinical endpoints is summarized in Table 3.

The convergence of these research priorities represents a foundational transition toward immune-centric cardiovascular medicine, offering the potential to transform outcomes for patients with diabetes through precision targeting of the immunological mechanisms contributing to accelerated atherosclerosis [136]. Success in this endeavor requires sustained commitment to interdisciplinary collaboration, innovative technological development, and rigorous clinical validation to realize the full therapeutic potential of immune checkpoint modulation in diabetic cardiovascular diseases.

### 6.10. Translational Barriers and Strategic Pathways for Clinical Implementation of Checkpoint-Based Therapies in Diabetic Atherosclerosis

Although preclinical evidence for checkpoint-based therapies in diabetic atherosclerosis is compelling, their translation into clinical practice faces substantial challenges [139]. An important challenge is to achieve therapeutic efficacy without inducing excessive immunosuppression. Oncological checkpoint blockade unleashes maximal immune responses. Cardiovascular checkpoint enhancement requires precise calibration. The goal shifts from activation to balanced immune regulation [140]. The therapeutic window between insufficient immune modulation and harmful suppression remains poorly defined, creating safety concerns regarding clinical implementation [141].

Preclinical studies with PD-L1 agonists and CTLA-4 mimetics demonstrated robust plaque reduction in diabetic mouse models; however, establishing a safe and effective dose in humans remains a major barrier, particularly in elderly patients already predisposed to infection and malignancy [142]. Systemic administration of checkpoint modulators risks widespread off-target effects while failing to achieve therapeutic concentrations in plaques [143]. Recent advances in nanotechnology offer potential solutions to these challenges. VCAM-1–targeted nanoparticles carrying PD-L1 agonists reduced lesion size by more than 60% in diabetic mice compared to untargeted delivery [144]. Similarly, drug-eluting stents capable of controlled release of CTLA-4 mimetics during revascularization procedures represent another promising approach; however, issues of drug stability, biocompatibility, and regulatory approval remain unresolved.

The clinical adoption of these methods also requires new management strategies. Unlike conventional cardiovascular therapies, checkpoint modulators necessitate specialized immune monitoring to detect early signs of excessive suppression while preserving vascular protection [145]. Although soluble PD-L1 and CTLA-4 levels show potential as biomarkers, standardized assays and reference ranges are not yet available [146]. Individuals with early subclinical disease may respond differently from those with advanced inflammatory plaques, underscoring the need for predictive biomarkers that integrate immune, genetic, and metabolic data [147].

Economic and regulatory barriers influence clinical feasibility as sophisticated nanoparticle-based delivery systems are costly to manufacture, and specialized infrastructure, such as cardio-immuno-metabolic clinics, will demand significant investment [148]. In addition, current regulatory frameworks developed for traditional cardiovascular therapies may not adequately address the safety and efficacy requirements of immune checkpoint modulation [149]. Post-COVID immune dysregulation intensifies checkpoint dysfunction, with SARS-CoV-2 infection causing persistent suppression of PD-1 and CTLA-4 and sustained endothelial injury. These abnormalities persist into long COVID, where checkpoint activity remains blunted and is accompanied by vascular inflammation and elevated cardiovascular biomarkers [150,151]. In patients with diabetes, this viral-induced checkpoint exhaustion compounds pre-existing deficiencies, producing synergistic immune–vascular dysregulation [152], contributing to disproportionate cardiovascular morbidity in post-COVID populations with metabolic diseases [153]. Similarly, chronic infections, such as human papillomavirus, have been linked to coronary artery disease through shared inflammatory and immune checkpoint pathways, further compounding cardiovascular risk [154], necessitating enhanced surveillance in this high-risk group. Despite these challenges, the therapeutic promise of checkpoint enhancement in diabetic atherosclerosis warrants continued investment, with success dependent on coordinated translational research, innovative delivery platforms, and rigorous safety evaluation [155].

### 6.11. Tissue-Resident Memory T Cells and Circadian Regulation the Unexplored Frontiers in Diabetic Atherosclerosis

Tissue-resident memory T cells (TRM) have emerged as key mediators of local immune surveillance, persisting at sites of chronic inflammation and adapting to the tissue microenvironment [156], often contributing to atherosclerotic plaque biology [157].

De Jong et al. identified a distinct CD69^+^CD49α^+^ TRM subset within human plaques that was associated with fibrous cap stability, greater collagen deposition and reduced macrophage infiltration, features consistent with protective remodeling [158]. This finding challenges the traditional view that all T cells within atherosclerotic lesions contribute to inflammation. The biology of TRM cells in diabetes remains unexplored, representing a major knowledge gap. Chronic exposure to hyperglycemia, advanced glycation end products, and pro-inflammatory cytokines may drive TRM exhaustion, marked by upregulation of inhibitory receptors, including PD-1, TIM-3, and LAG-3, thereby weakening protective functions [159]. Alternatively, metabolic stress in diabetic tissues may skew TRM cells toward pro-inflammatory phenotypes through enhanced glycolysis and dysregulated lipid metabolism, amplifying cytokine-driven damage [160].

Therapeutically, these divergent pathways have distinct implications for treatment strategies. If exhaustion dominates, PD-1/PD-L1 modulation or metabolic interventions may restore protective TRM activity [161]. If pro-inflammatory skewing prevails, depletion or reprogramming strategies may be required. The localized nature of TRM cells makes them attractive for targeted delivery approaches that minimize systemic immunosuppression [162]. Future studies employing single-cell RNA sequencing and spatial transcriptomics will be critical to define the metabolic and transcriptional programs that shape TRM fate in diabetic atherosclerosis [163]. In parallel, microRNA-based regulation of lipid metabolism and inflammatory pathways particularly through dual targeting of miR-33 and miR-92a, has emerged as a promising strategy to modulate atherogenesis, offering potential synergy with checkpoint-based therapies in addressing the metabolic–immune axis of diabetic vascular disease [164].

Notably immune checkpoint pathways are under circadian regulation, introducing a temporal dimension to therapeutic efficacy. Fortin et al. demonstrated that PD-L1 expression on myeloid cells oscillates with daily rhythms, peaking during active phases via regulation by core clock factors such as CLOCK and BMAL1 at the CD274 promoter [165]. This suggests that the timing of checkpoint-targeted therapy may critically influence outcomes.

Circadian regulation is profoundly disrupted in diabetes. Altered sleep, feeding patterns, and hormone cycles, along with molecular clock dysfunction within immune cells, reduce checkpoint integrity [166]. Hyperglycemia interferes with clock gene expression, and inflammation uncouples immune oscillations from systemic rhythms [167]. These disruptions affect checkpoint pathways through metabolic–epigenetic crosstalk with clock-regulated NAD^+^ production modulates SIRT1 activity at PDCD1, while core clock genes influence PD-1 transcription during T cell activation [168]. These findings establish a rationale for circadian-informed checkpoint therapy in diabetes. Chronotherapy principles, including time-specific dosing, timed feeding, and light-based interventions, may help restore checkpoint oscillations [169]. Adjunctive strategies such as time-restricted feeding, melatonin supplementation, or pharmacological modulators of clock pathways (e.g., REV-ERB agonists) could normalize circadian and immune function [170]. Clinical translation will require biomarker-driven trials that incorporate chronotype assessment and serial checkpoint profiling to align interventions with optimal circadian phases. Evidence shows that checkpoint integrity can be measured using circulating and tissue biomarkers. For example, soluble PD-1 and PD-L1 (sPD-1, sPD-L1) in plasma correlate with T-cell exhaustion and cardiovascular risk [171]. Flow cytometric analysis of PD-1, CTLA-4, TIM-3, and LAG-3 on CD4^+^ and CD8^+^ T cells provide a direct assessment of checkpoint activity and immune exhaustion [172]. Single-cell transcriptional profiles, such as high PDCD1 expression, co-expression of TIGIT and LAG-3, and reduced IL-7R, as well as plasma markers such as cytokine ratios (IL-6/TGF-β), metabolic indicators (lactate, mitochondrial membrane potential), and epigenetic exhaustion features provide additional quantitative readouts of checkpoint dysfunction [173]

The hypothesis that circadian timing modulates vascular TRM biology and responsiveness to checkpoint restoration can be evaluated using a staged translational strategy. In the preclinical phase, diabetic ApoE−/− or LDLR−/− mice may be exposed to controlled or disrupted light–dark cycles, with checkpoint agonist administration delivered during either the rest or active phase. The circadian control of T-cell activity has been demonstrated in several models, including BMAL1-dependent regulation of monocyte and T-cell rhythms [174]. Complementary studies using T-cell specific BMAL1 deletion can test whether core clock disruption alters TRM responsiveness, supported by evidence that BMAL1 signaling in CD4^+^ T cell shapes immune activation [175]. The TRM-specific endpoints such as CD69, CD103 and CXCR6 expression, checkpoint receptor levels and single-cell transcriptomic signatures are grounded in established TRM biology [176]. A translational human step may analyze carotid or coronary tissue collected across different times of day to correlate TRM and clock gene signatures. Finally, a randomized crossover pilot study in patients with stable atherosclerosis could compare morning versus evening dosing of a checkpoint-modulatory agent, with vascular inflammation monitored using FDG-PET, an established marker of plaque activity [177]. This sequence of mechanistic, ex vivo, and early clinical approaches provides an ethically feasible and scientifically rigorous framework for testing the circadian–TRM–checkpoint axis.

Integrating TRM biology and circadian regulation into therapeutic design marks a fundamental shift toward precise, tissue- and time-targeted immunomodulation, moving beyond systemic approaches to achieve a more focused checkpoint restoration.

### 6.12. Checkpoint Dysfunction in Type 1 Diabetes and Cardiovascular Risk

Type 1 diabetes (T1D) arises from the autoimmune destruction of pancreatic β-cells driven by early failures in central and peripheral tolerance, which implicate checkpoint pathways. Pancreatic β-cells express PD-L1 as a protective mechanism against autoreactive T cells, and blockade or genetic loss of PD-1/PD-L1 precipitates or accelerates autoimmune diabetes in preclinical models [78]. Human studies have shown PD-L1 expression in residual insulin-positive islets and linked checkpoint pathway disruption to ICI-associated insulin-dependent diabetes in patients receiving anti-PD-1/PD-L1 therapy [178]. Genetic variation in immune regulatory loci, including CTLA4 and PDCD1, modifies T1D susceptibility and may influence responses to checkpoint-targeted interventions [179].

Checkpoint failure in T1D is not restricted to pancreatic islets but extends systemically, contributing to early onset vascular dysfunction. Subclinical atherosclerotic changes, such as increased carotid intima–media thickness, are observed in children and young adults with T1D [180]. Mechanistically, autoimmune checkpoint deficiency combines metabolic and epigenetic alterations to sustain suppression of PD-1/PD-L1 and related regulators across multiple tissues, amplifying vascular inflammation and plaque vulnerability. Key metabolic and chromatin factors include altered α-ketoglutarate/succinate ratios, NAD^+^ depletion, and persistent histone and DNA modifications that impair TET and sirtuin function [52].

While systemic checkpoint dysfunction in T1D increases the risk of vascular disease, the early onset of the disease provides a scope for preventive vascular interventions. Early-phase studies should prioritize vascular-targeted delivery or local modulation, recruit patients with recent-onset disease and residual C-peptide and incorporate serial immune and metabolic phenotyping to ensure safe and informative clinical translation.

### 6.13. Translational Priorities and Research Roadmap for Checkpoint Restoration in Diabetes

Advancing checkpoint restoration into clinical practice requires a coordinated framework that integrates preclinical precision with clinical translation.

Prospective studies with detailed immune profiling are essential to determine whether checkpoint expression at diagnosis or during prediabetes can predict cardiovascular outcomes beyond established risk factors. In parallel, systematic dose-finding and safety studies in diabetic models are required to define minimally effective exposures that restore vascular tolerance without impairing antimicrobial defense or tumor immunosurveillance.

Comparative studies of human vascular tissues, using single-cell RNA sequencing and spatial transcriptomics, should clarify vessel-specific checkpoint regulation across coronary, carotid, and peripheral beds and identify pharmacodynamic markers with cell-type specificity. Because metabolic and epigenetic programs stabilize checkpoint suppression, therapeutic strategies must combine metabolic remodeling with targeted immune modulation. Potential approaches include NAD^+^ restoration, metabolic epigenetic modifiers that sustain TET and sirtuin activity, and vascular-targeted delivery systems that limit systemic immune effects [181]. The immunomodulatory influence of widely used cardiometabolic drugs such as metformin, statins, SGLT2 inhibitors and GLP-1 receptor agonists remains insufficiently defined and demands mechanistic and translational evaluation [182].

The design of early human trials should be biomarker-driven and adaptive. Deep baseline immune and genetic profiling will enable pharmacogenomic stratification, while imaging and circulating checkpoint markers can serve as intermediate endpoints. Infection and tumor surveillance must be built in as safety safeguards. Integration of multi-omic datasets with advanced computational models offers the possibility of identifying high-risk phenotypes, predicting therapeutic response and refining trial enrichment, but such models require rigorous external validation and close attention to bias before clinical use.

### 6.14. Predictive Analytics and AI for Personalizing Checkpoint-Based Cardiovascular Therapies

The complexity of immune checkpoint dysfunction in diabetes demands approaches beyond conventional clinical assessment. Yang et al. recently demonstrated the utility of predictive algorithms by identifying cancer patients at risk of hyperglycemia during PD-1 therapy, highlighting the promise of computational models in immunotherapy [183]. A similar framework could transform cardiovascular applications, where checkpoint disruption reflects a convergence of metabolic, circadian, epigenetic, and pharmacological perturbations.

Traditional measures such as glycemic indices are insufficient to capture this multidimensional dysfunction. Machine learning can integrate diverse datasets on immune signatures, genetic variants, and temporal dynamics to generate individualized risk profiles [183,184].

Emerging “immune risk scores,” which combine checkpoint expression, inflammatory biomarkers, and genetic susceptibility, illustrate this potential [184]. Such tools could identify patients with suppressed PD-1 signaling and elevated cytokines who would benefit from checkpoint modulation, while sparing those with preserved function from unnecessary therapy. Beyond patient selection, AI-guided approaches could improve trial design by enriching likely responders, reducing required sample sizes, and accelerating drug development [185].

Incorporating multi-omics platforms, transcriptomic, metabolomic, and proteomic profiling will be essential for refining predictions [186]. However, translation into clinical practice requires rigorous validation across populations, regulatory alignment, and strategies to mitigate algorithmic bias, particularly in diabetes care [187]. Overcoming these barriers is critical for realizing checkpoint-based precision medicine for cardiovascular diseases.

Experimental studies have shown that PD-1/PD-L1 activation following myocardial infarction exerts a cardioprotective role by containing excessive inflammatory responses during reperfusion [188]. The upregulation of PD-L1 on cardiac myocytes and infiltrating immune cells reflects an adaptive mechanism that minimizes collateral injury while preserving antimicrobial defenses. However, in diabetes, checkpoint pathways are impaired, limiting the protective effect. Such deficiencies may intensify post-infarction inflammation, prolong tissue injury, and impede recovery [189], potentially explaining the higher complication and mortality rates in patients with diabetes after acute coronary syndromes. Comparable mechanisms are implicated in acute stroke, where impaired checkpoint regulation permits unchecked microglial activation and neuroinflammation, contributing to larger infarct volumes and poorer outcomes [190].

### 6.15. Therapeutic Interactions and Molecular Crosstalk

Despite significant progress in understanding immune checkpoint dysfunction in diabetic atherosclerosis, several critical gaps hinder translation into clinical practice. A major unresolved issue is whether quantitative checkpoint expression profiling at the time of diabetes diagnosis can independently predict cardiovascular risk beyond established biomarkers. This question can only be answered by prospective cohort studies with serial immune phenotyping and long-term follow-up. Equally important is defining the therapeutic window for checkpoint agonists. The minimal effective dose that restores vascular tolerance without compromising antimicrobial defense or tumor immunosurveillance remains unknown, necessitating dose-escalation studies in relevant diabetic models with rigorous safety monitoring.

Checkpoint regulation may also be tissue specific. Coronary, carotid, and peripheral vessels display distinct patterns of expression, suggesting that anatomically tailored approaches may be necessary. Comparative human studies employing single cell sequencing and spatial transcriptomics are urgently required to address this gap. Another unresolved question is the reversibility of checkpoint dysfunction with intensive glycemic control. Longitudinal biomarker monitoring is needed to distinguish between true restoration of immune tolerance and mere arrest of ongoing deterioration.

Genetic factors add another layer of complexity to this issue. Variants in PDCD1, CD274, and CTLA4 are likely to shape individual susceptibility, and the incorporation of these variants into polygenic risk scores could enable more precise patient stratification. The impact of commonly used diabetes and cardiovascular therapies, such as metformin, statins, SGLT2 inhibitors, and GLP-1 receptor agonists, on checkpoint signaling remains insufficiently understood, and clarifying these interactions is essential for developing rational therapeutic combinations. Equally critical is a deeper understanding of the molecular crosstalk between metabolic stress pathways and checkpoint regulation at the transcriptional, epigenetic, and post-translational levels, which may uncover novel targets for intervention. Mapping these interactions may uncover druggable nodes with therapeutic potential. Whether checkpoint dysfunction precedes clinical hyperglycemia in prediabetes is another key question that could create a window for preventive intervention before irreversible vascular damage occurs. Furthermore, the possibility that checkpoint molecules may serve as surrogate endpoints in cardiovascular outcome trials has not been explored and such biomarkers could accelerate drug development and regulatory approval.

Finally, the complexity of immune–metabolic interaction calls for advanced integrative approaches. Artificial intelligence applied to multi-omics datasets could provide powerful tools for predicting individual disease trajectories and therapeutic responses, thereby advancing precision medicine in diabetic vascular disease. Addressing these gaps through coordinated mechanistic research, longitudinal cohort studies, and adaptive trial designs will determine whether checkpoint restoration can move beyond experimental promise to become an evidence-based strategy capable of altering the natural history of diabetic vascular diseases.

## 7. Conclusions

Immune checkpoint dysfunction represents a central mechanistic link between metabolic dysregulation and accelerated atherosclerosis in diabetes. Hyperglycemia-driven suppression of the PD-1/PD-L1 and CTLA-4 pathways leads to persistent T cell activation, regulatory T cell failure, and endothelial barrier instability, creating an ideal environment for vascular inflammation and plaque progression. Although some studies have reported partial improvement in checkpoint expression with improved glycemic control, this recovery is inconsistent because short-term metabolic normalization cannot undo the long-standing epigenetic repression of PD-1/PD-L1 and CTLA-4 loci. Thus, glycemic control can stop further deterioration of checkpoint function but does not typically reverse the long-standing metabolic–epigenetic suppression caused by chronic hyperglycemia, which explains the conflicting findings reported in different studies. Integrating the immune checkpoint biomarkers with metabolic and genetic indices enables the development of checkpoint-informed risk panels that can identify patients at elevated cardiovascular risk beyond conventional glycemic or lipid parameters. In parallel, molecular imaging of immunologically active plaques using PD-L1–targeted PET combined with coronary CT angiography or cardiac MRI offers the possibility of detecting subclinical, rupture-prone lesions, converting vascular assessment from static anatomic evaluation to dynamic immune surveillance. Early phase clinical trials of vascular-targeted checkpoint-restoring interventions, designed with serial immune phenotyping, advanced plaque imaging, and rigorous safety monitoring, are essential to define therapeutic windows and establish proof-of-concept efficacy of these therapies. The coordinated implementation of these strategies within specialized cardio-immuno-metabolic platforms could not only refine risk stratification and enable early detection but also provide a framework for immune-guided therapies to prevent or reverse diabetic atherosclerosis. By translating mechanistic insights into actionable clinical pathways, the modulation of immune checkpoints offers a tangible opportunity to improve cardiovascular outcomes in millions of individuals living with diabetes worldwide.

## Figures and Tables

**Figure 1 biology-14-01731-f001:**
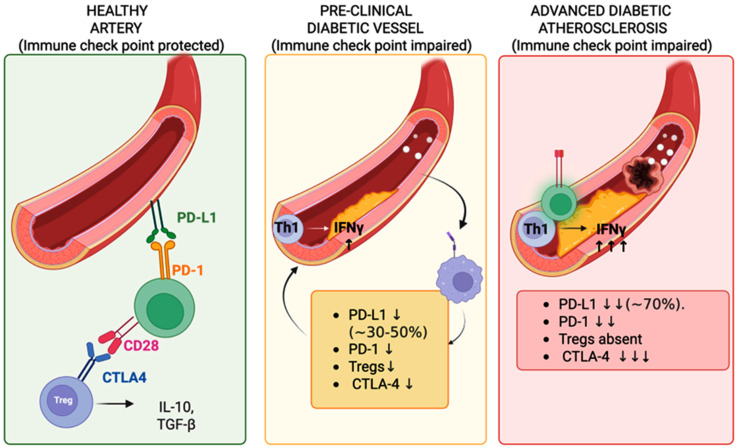
Progressive impairment of vascular immune checkpoints in diabetes. In healthy arteries, immune equilibrium is maintained through the signaling pathways of PD-L1, PD-1, and CTLA-4, in conjunction with cytokines derived from regulatory T cells (Tregs). In pre-diabetic vessels, there is a partial loss of these checkpoints, a reduction in Tregs, and an increase in Th1-driven interferon-gamma (IFN-γ), which facilitates the early formation of plaques. In the advanced stages of the disease, there is a near-complete loss of checkpoint function, absence of Tregs, and heightened inflammation, culminating in the development of unstable plaques. (Created with BioRender.com).

**Figure 2 biology-14-01731-f002:**
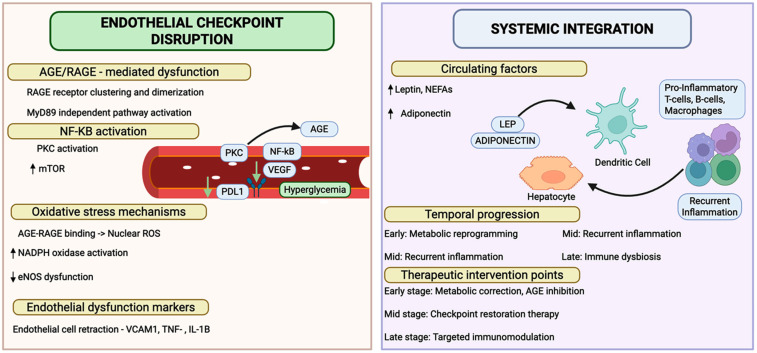
Endothelial checkpoint disruption and systemic immune integration in diabetes. Hyperglycemia initiates the activation of AGE–RAGE signaling, NF-κB, and oxidative stress pathways, resulting in diminished PD-L1 expression and compromised immune regulation in the endothelium. Activation of PKC and mTOR further exacerbates these effects, whereas dysfunctional endothelial cells exhibit markers such as VCAM-1, TNF, and IL-1β. Systemic alterations, including changes in leptin, NEFA, and adiponectin levels, impair dendritic cell function and facilitate the chronic activation of T and B cells, thereby perpetuating recurrent inflammation. The disease trajectory progresses from initial metabolic reprogramming to chronic inflammation and immune imbalance. Potential intervention points are identified at the levels of metabolic correction, checkpoint restoration, and targeted immunomodulation. (Created with BioRender.com).

**Figure 3 biology-14-01731-f003:**
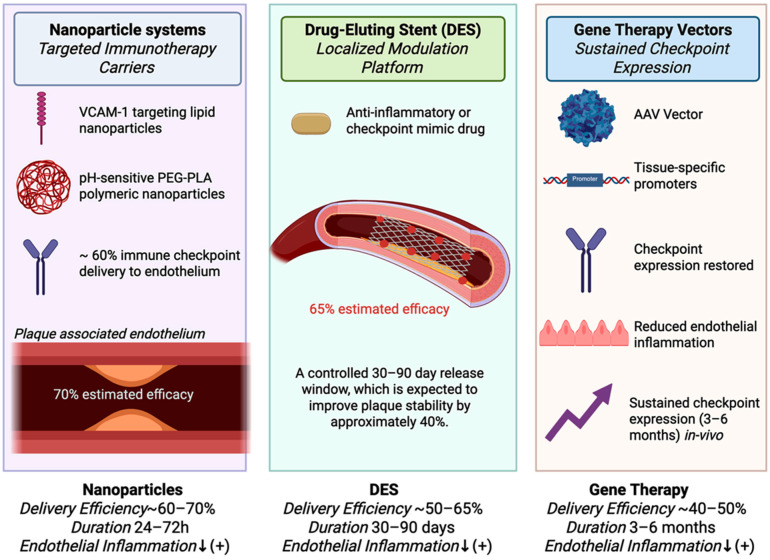
Therapeutic strategies for restoring endothelial checkpoint function. This figure compares three approaches used to modulate vascular immune checkpoint signaling. On the (**left**), nanoparticle systems such as VCAM-1–targeted lipid particles and pH-sensitive polymeric nanoparticles provide short-term but highly targeted checkpoint delivery to inflamed endothelium. In the (**center**), drug-eluting stents (DES) release anti-inflammatory or checkpoint-mimetic agents over a 30–90-day period, improving local plaque stability and reducing vascular inflammation. On the (**right**), gene therapy vectors use tissue-specific promoters to maintain checkpoint expression for several months in vivo, leading to sustained reductions in endothelial inflammation even with modest delivery efficiency. (Created with BioRender.com).

**Table 1 biology-14-01731-t001:** Evidence Linking Immune Checkpoint Dysfunction to Cardiovascular Risk in Diabetes.

Study Category	Key Findings	Clinical Insight	Translational Gap	Reference
Immune Checkpoint Blockade (ICI) and Cardiovascular (CV) Risk	ICIs increase CV events 3.4-fold	Validates that immune checkpoints serve protective roles in vasculature	Cancer populations differ from diabetics, unclear if same risk mechanisms apply directly in metabolic disease	[75]
T Cell Phenotyping in Diabetes	Diabetics show suppressed PD-1/CTLA-4 expression on T cells	Direct link between immune checkpoint impairment and atherosclerotic risk in diabetic patients	No longitudinal data to confirm if this dysfunction precedes or predicts CV events	[76]
Effect of Glycemic Control	Poor glucose control (HbA1c > 8.5%) associates with lower PD-1 on effector T cells	Dose–response between hyperglycemia and immune dysregulation	Lacks interventional evidence that improving glycemia restores checkpoint integrity or improves outcomes	[77]
Endothelial Checkpoint Expression	70% reduction in PD-L1 in diabetic endothelium	Suggests vascular tissue participates in immune dysregulation	Functional impact on endothelial-immune cross-talk remains speculative without in vivo confirmation	[78]
Treg Checkpoint Functionality	Tregs exhibit a 60% reduction in CTLA-4 expression in diabetic patients	Highlights dysfunction in regulatory arms of immune tolerance	No clinical trials have evaluated whether restoring Treg function alters CV risk in diabetes	[79]
Experimental Therapeutic Modulation	Pilot data show that checkpoint restoration reduces inflammation and improves endothelial function	Suggests reversibility and therapeutic targetability of immune dysfunction in diabetes	Limited sample and short duration, with uncertain effects on hard CV outcomes	[80]

**Table 2 biology-14-01731-t002:** Comparative roles of major immune checkpoints in vascular homeostasis, their dysregulation in diabetes, and emerging therapeutic strategies.

Checkpoint	Key Functions in Vasculature	Effect of Diabetes	Therapeutic Strategy
PD-1/PD-L1 [76]	Suppresses T cell activity, stabilizes plaques	Downregulated in diabetes, increased CD8^+^ infiltration	PD-L1 agonists, gene upregulation
CTLA-4 [77]	Supports Treg suppression of inflammation	Impaired function, reduced expression	CTLA-4 mimetics, Treg-based therapies
LAG-3, TIM-3, TIGIT [78]	Emerging regulatory roles in atherogenesis	Poorly characterized	Future targets for intervention

**Table 3 biology-14-01731-t003:** Proposed Clinical Trial Design for Immune Checkpoint-Based Therapy in Diabetic Atherosclerosis.

Parameter	Design Consideration
Population	T2DM patients with subclinical or symptomatic atherosclerosis
Intervention	PD-L1 agonist (e.g., mAb, nanoparticle) or CTLA-4 mimetic
Control	Placebo or standard care
Primary Outcome	MACE (MI, stroke, CV death)
Secondary Outcomes	hsCRP, IL-6, T cell activation, plaque composition (imaging)
Biomarkers	sPD-L1, CD8^+^/PD-1^+^ ratios, Treg/Th17 balance

## Data Availability

No new data were created or analyzed in this study.

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
