# Peer review of "Immune Checkpoint Restoration as a Therapeutic Strategy to Halt Diabetes-Driven Atherosclerosis"

_biology, 2025, doi:10.3390/biology14121731_

Round 1
Reviewer 1 Report
Comments and Suggestions for Authors
Suggestion: Major Revision
This manuscript presents an extensive and conceptually rich review on immune checkpoint dysfunction as a therapeutic target in diabetes-driven atherosclerosis. The mechanistic integration of metabolic, epigenetic, and immunologic pathways is impressive, but the review would benefit from clearer data synthesis, reduced redundancy, and sharper translational focus. Overall, it is a valuable contribution but requires structural refinement and concise highlighting of clinical implications.
Review Questions
1. How does the proposed framework differentiate causality from correlation in linking checkpoint dysfunction with diabetic atherosclerosis?
2. Are the cited preclinical and clinical studies sufficiently robust to justify checkpoint restoration as a viable therapeutic strategy?
3. Can the authors clarify how PD-1/PD-L1 agonism avoids systemic immunosuppression, especially in diabetic populations with compromised immunity?
4. What quantitative criteria or biomarkers are proposed for assessing checkpoint integrity in clinical settings?
5. How does the review reconcile conflicting evidence on the impact of glycemic control on immune checkpoint recovery?
6. Could the authors provide more specific discussion on safety considerations in vascular-targeted delivery systems and their pharmacokinetic validation?
7. How might individual genetic variability (e.g., PDCD1, CTLA4 polymorphisms) influence therapeutic response or patient selection?
8. What evidence supports the feasibility of combining checkpoint restoration with standard antidiabetic or lipid-lowering therapies?
9. The section on circadian regulation and TRM cells introduces novel ideas can the authors suggest a feasible experimental or clinical design to test these hypotheses?
Reviewer 2 Report
Comments and Suggestions for Authors
Author Recommendations:
Study Design and Result Interpretation:
-
The paper is conceptually strong but would benefit from a clearer delineation between established evidence and speculative hypotheses. Some mechanistic discussions (e.g., epigenetic remodeling and checkpoint modulation) should be labeled as theoretical rather than empirically validated.
-
The authors discuss potential clinical translation but should temper conclusions about therapeutic feasibility until supported by clinical data.
Methods/Discussion Consistency:
-
The structure is consistent, but several sections overlap conceptually (Sections 3.1 to 3.3 on checkpoint suppression mechanisms). These could be merged or shortened for clarity.
-
The figures, while visually strong, could benefit from explicit referencing within the text where each concept is first introduced
Minor Editorial Suggestions
References:
-
References are current and appropriate, though the authors could include recent 2024–2025 immunometabolism studies to contextualize checkpoint restoration beyond diabetes (e.g., Nature Metabolism and Circulation Research papers on immune-metabolic crosstalk).
-
Some references (e.g., [71], [72]) are cited multiple times for overlapping content and could use a little condensing for more concise readability
Data Presentation:
-
Tables 1 to 3 are informative but lengthy; condense repetitive rows and add concise explanatory legends.
-
Figures 1 to 3 could use larger labels and simplified annotations for readability.
Readability and Structure:
-
The manuscript is highly technical. Some long sentences could be simplified into shorter clauses to aid comprehension.
-
Consider adding brief “Summary” sentences at the end of each section to reinforce key points and summarize the preceding paragraphs.
-
Some paragraphs (e.g., Section 6.10–6.16) could be tightened by combining overlapping themes.
Final Comments:
This is an ambitious, deeply researched, and conceptually rich review that bridges immunology and cardiometabolic disease. The topic is timely and well aligned with the Biology journal’s scope. To enhance readability and scientific precision, I recommend:
-
Simplifying the prose and minimizing redundancy between mechanistic sections.
-
Adding brief clarifying statements to distinguish hypotheses from proven mechanisms.
-
Expanding the discussion of potential risks of checkpoint restoration, especially tumor surveillance and infection vulnerability.
-
Shortening the “Future Directions” section to focus on the most actionable translational strategies.
-
Adding a graphical abstract summarizing the central concept: immune checkpoint restoration as a link between diabetes and atherosclerosis.
Overall, this is a high-quality manuscript with strong translational potential that requires moderate refinement for clarity and precision.
Reviewer 3 Report
Comments and Suggestions for Authors
This review article synthesized evidence linking immune checkpoint dysfunction (especially PD-1/PD-L1 and CTLA-4 pathways) to diabetes-driven atherosclerosis, and it proposed immune checkpoint restoration as a therapeutic strategy to prevent or slow vascular disease progression. This manuscript integrates metabolic and immune mechanisms to explain how hyperglycemia disrupts major immune checkpoint signaling, driving chronic vascular inflammation in diabetes. Importantly, it provided evidence from clinical studies to depict that loss of checkpoint control accelerates atherosclerosis.
This article is interesting and well written. It presents a comprehensive overview of immune checkpoint pathways in diabetes-driven atherosclerosis, which is of major interest in the field.
I have a few comments.
- I suggest that the authors label all figure panels sequentially (A, B, C, etc.) and ensure that these labels are clearly referenced and explained in the corresponding figure legends.
- I observed that numerous abbreviations are used throughout the manuscript. I recommend that the authors include a comprehensive list of all abbreviations at the end of the manuscript.
Round 2
Reviewer 1 Report
Comments and Suggestions for Authors
fter careful examination and evaluation the manuscript may be accepted in it's current form.